# Measuring the molecular origins of stiffness in organic semiconductors

Ki-Hwan Hwang[1,2] ✉, Dorothée Brandt[3], Silvia Cristofaro[3], Cameron J. Nickerson[4], Federico Modesti[5], Mindaugas Gicevičius[1], Mateo T. R. Cervantes[1], Martina Volpi ⓘ [6], Leszek J. Spalek[1,7], Luca Muccioli[8], Per M. Claesson ⓘ [2], Ljiljana Fruk[7], Yves Geerts[6,9], Guillaume Schweicher[6], Yoann Olivier ⓘ [3], Erin R. Johnson[4,10,11] & Deepak Venkateshvaran ⓘ [1,7,12] ✉

Mechanical properties of organic molecular semiconductors are determined by a combination of chemical structure and solid-state packing. Measurements of nanoscale mechanical properties on molecular surfaces via atomic force microscopy (AFM) are particularly challenging as the very act of probing how stiff these surfaces are may perturb them, making it difficult to discern subtle differences in stiffness arising from changes in molecular composition. This work presents the first direct, experimental demonstration of the tunability in the nanomechanical properties for a family of molecular semiconductors resulting from systematic alkyl sidechain substitution. While such tunability is intuitively expected, it is a subtle effect that is extremely difficult to detect and quantify reliably from nanoscale AFM measurements due to various spurious force contributions operating on such small length scales. Only after identifying and removing these spurious contributions is the underlying molecular-scale tailoring of mechanical properties observable. Confidence in the measured stiffness trend is reinforced through simulations based on density-functional theory (DFT) and molecular dynamics (MD).

For several decades, academic and industrial research on organic semiconductors has focused on their optical properties, electronic properties, and thermal properties. Applications that stem from such research include solar and thermal energy harvesting, bioelectronics, as well as new characterisation techniques[1–7]. Organic semiconductors are known for their large-area flexibility and macroscopic bendability; unique selling points central to most applications. Although their macroscopic flexibility is taken for granted, their mechanical stiffness on the micro and nano scale is only beginning to be investigated, thanks to the emergence of new experimental measurement techniques[8].

Nanomechanical measurements are carried out using specific modes on atomic force microscopes that capture the complete time-extended force interaction between the cantilever tip and the surface under study. This force-distance curve quantifies the total tip-surface

[1]Cavendish Laboratory, University of Cambridge, JJ Thomson Avenue, Cambridge, United Kingdom. [2]KTH Royal Institute of Technology, School of Engineering Sciences in Chemistry, Biotechnology and Health, Department of Chemistry, Division of Surface and Corrosion Science, Stockholm, SE-100 44, Sweden. [3]Laboratory for Computational Modeling of Functional Materials, Namur Institute of Structured Matter, University of Namur, Namur, Belgium. [4]Department of Physics and Atmospheric Science, Dalhousie University, Halifax, Canada. [5]BASF SE, Ludwigshafen am Rhein, Germany. [6]Laboratoire de Chimie des Polymères, Faculté des Sciences, Université Libre de Bruxelles (ULB), CP 206/01, Brussels, Belgium. [7]Department of Chemical Engineering and Biotechnology, University of Cambridge, Philippa Fawcett Drive, Cambridge, United Kingdom. [8]Department of Industrial Chemistry "Toso Montanari", University of Bologna, Bologna, Italy. [9]International Solvay Institutes of Physics and Chemistry, Université Libre de Bruxelles (ULB), CP 231, Brussels, Belgium. [10]Yusuf Hamied Department of Chemistry, University of Cambridge, Cambridge, United Kingdom. [11]Department of Chemistry, Dalhousie University, Halifax, Nova Scotia, Canada. [12]Department of Science Innovation and Technology, UK Government, 100 Parliament Street, London, SW1A 2BQ, United Kingdom. ✉e-mail: kh826@cam.ac.uk; dv246@cam.ac.uk

interaction in stages: (a) as the cantilever is driven towards the surface, (b) as it is brought into contact with the surface, (c) as it makes an elastic deformation to the top few nanometres of the surface, and (d) as it is retracted to large separation where the force interaction is zero[9,10]. Several properties can be acquired through the measured force-distance curves, including surface adhesion, surface deformation, indentation depth, nanoscale stiffness and, within reasonable model-based estimation, the Young's modulus. Macroscopically, stiffness represents a resistance to deformation under an external force and is an ensemble property of molecules under equilibrium.

Together with the wealth of existing knowledge on charge-carrier mobilities in organic semiconductors, precision measurements of stiffness enable a deeper understanding of how their electronic properties and mechanical properties correlate. Generalisations from time-tested theories that link the charge-carrier mobility with the elastic modulus, such as Bardeen and Shockley's deformation potential theory published originally in 1950, may ultimately be deployed to understand the link between electronic properties and elastic properties in appropriately oriented high-mobility organic semiconductors[11].

Nanomechanical measurements of molecular thin films are non-trivial and require significant attention to detail. Unlike typical optical,

electronic, or thermal measurements of molecular semiconductors that do not involve contact-based surface deformation, nanomechanical measurements do. The very act of probing a molecular surface mechanically can perturb it, leading to inconsistent measurement estimates.

In this work, the link between nanoscale stiffness, studied through force-distance curve mapping on an AFM, and the chemical structure of the organic molecules that compose a lattice is explored. Specifically, we conduct a comparative study of nanoscale stiffness for a family of organic small molecule semiconductors based on the dinaphtho[2,3-b:2',3'-f]thieno[3,2-b]thiophene (DNTT) aromatic core. Artefacts that arise when performing AFM-based nanomechanical measurements are discussed, including the hidden nuances of nanoscale tip-sample interactions that are generally overlooked in routine measurements. Under careful measurement conditions, it becomes possible to discern minute differences in nanoscale mechanical properties of this family of materials. Mechanical properties predicted using molecular dynamics and density-functional theory show consonance with experiment. Finally, we conjecture as to how mechanical properties and electronic properties may be connected under specific conditions.

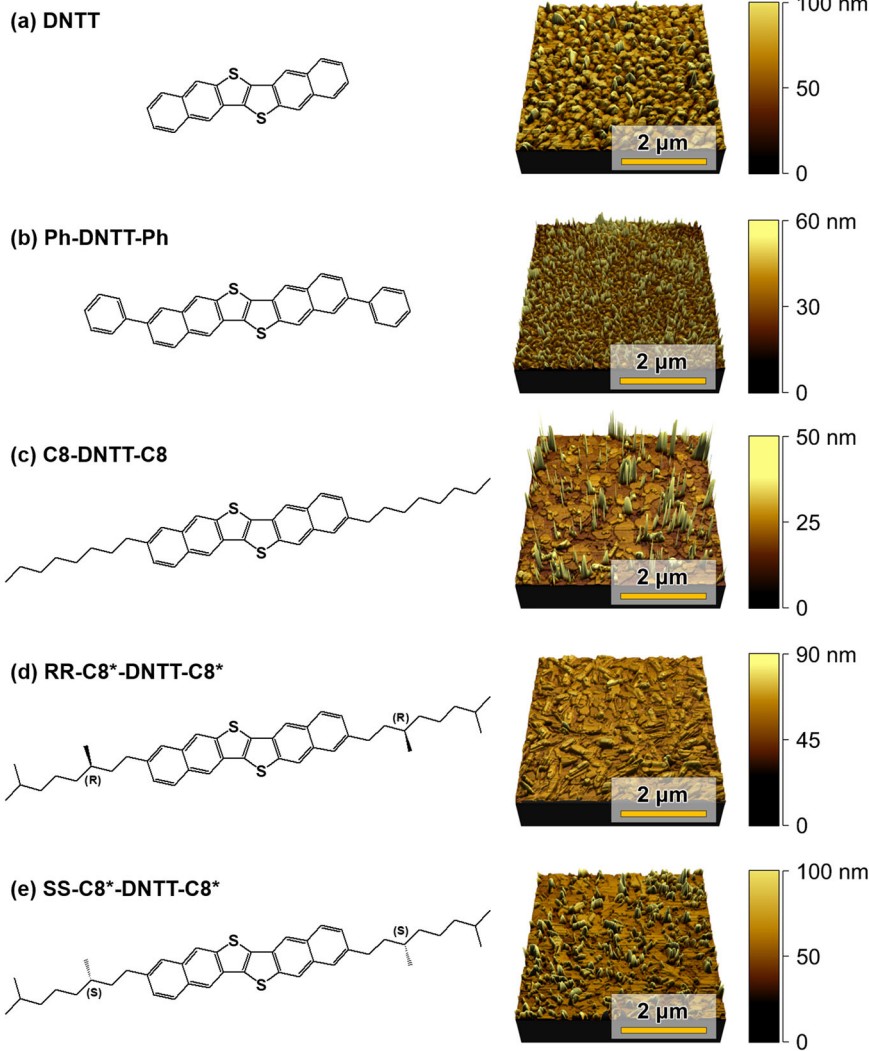

**Fig. 1 | Chemical structures and surface topography of DNTT and its derivatives. a–e** show the chemical structures and corresponding nanoscale topography of DNTT, Ph-DNTT-Ph, C8-DNTT-C8, RR-C8*-DNTT-C8*, and SS-C8*-DNTT-C8* thin films deposited by thermal evaporation. The films were grown at temperatures optimised for achieving the highest mobility in thin-film transistors fabricated from these materials (80, 90, 100, 40 and 40 °C, respectively).

## Results

### Chemical structure and surface topography of DNTT and its derivatives

Figure 1 shows the organic molecules based on the DNTT structural unit together with their accompanying surface textures. The thin film samples were measured in the Quantitative Imaging (QI) mode of a JPK atomic force microscope (See Methods)[12–15]. The measured surface topographies were confirmed on a separate set of samples investigated on a Park Systems NX10 atomic force microscope. Besides unsubstituted DNTT, shown in Fig. 1a, we studied a phenyl disubstituted DNTT derivative (Ph-DNTT-Ph, Fig. 1b, sometimes called diPh-DNTT in the literature),[16,17], DNTT with octyl sidechains (C8-DNTT-C8, Fig. 1c[18], and two DNTT derivatives bearing chiral centres on their side chains (Fig. 1d, e). In this work, we refer to the latter as RR-C8*-DNTT-C8* (sometimes called R-DNTT in the literature), and SS-C8*-DNTT-C8* (sometimes called S-DNTT in the literature)[19]. These molecular semiconductors were deposited using thermal evaporation (see Methods) at base substrate temperatures optimised for the highest charge-carrier mobilities in polycrystalline thin-film transistor devices (see caption of Fig. 1). It is clear from the surface topographies that each film's growth is mediated by different modes, namely, Volmer-Weber or island formation as seen predominantly for Ph-DNTT-Ph, Frank-van der Merwe or layer-by-layer growth as seen predominantly for DNTT, Stranski-Krastanov or layer-plus-island growth seen in C8-DNTT-C8 (more islands and fewer terraces), SS-C8*DNTT-C8* (more terraces and fewer islands), and RR-C8*-DNTT-C8* (a hybrid between terraces and islands)[20]. It is known from ultrathin films of DNTT (a couple of nanometres thin) that continuous surface reorganisation takes place due to molecular migration up to several hours after the films have been deposited[21,22]. We have found that such surface reorganisation takes place on our films too, even though they are tens of nanometres in thickness. For this reason, we chose to probe the surface topographies of DNTT and all its derivatives shown in Fig. 1 a few months after deposition. During this time, the samples were left sealed in vacuum packs, at room temperature, and were never exposed to ambient conditions from the time of their deposition. Right before measuring, each sample was removed from its vacuum pack and measured within 20 min of ambient exposure. Repeat measurements performed days later confirmed that there was no change in the surface topography. In what follows, we compare measurements of the nanoscale stiffness for these five organic molecular surfaces, optimised for charge-carrier mobilities, and demonstrate the impact of side-chain functionalisation on their nanomechanical properties.

### Measurement of nanomechanics in DNTT and its derivatives

The nanomechanical properties of DNTT and its family of derivatives were measured using specific nanomechanical modes on an atomic force microscope (see Methods). This technique is fast becoming a way to probe the stiffness of materials on the nanoscale, with resolutions down to a couple of nanometres[23,24]. Figure 2a shows a schematic diagram of the cantilever together with its apex and demonstrates how the cantilever moves from one point to the next during measurement. The cantilever approaches the surface (along the line shown in blue), depresses the surface to a couple of nanometres, and retracts from the surface (along the line shown in red). The cantilever then moves to an adjacent pixel, and the same measurement is repeated. Upon contact with the surface, the apex of the rounded cantilever's probe makes a viscoelastic deformation of the organic semiconductor surface, shown in Fig. 2b. The force is determined by the setpoint on the atomic force microscope, typically between 5 and 100 nN[24–27].

The force-distance curve, necessary to extract the stiffness, is first mapped as a force-time curve shown in Fig. 2c. Considering the bending of the cantilever and the height of the z-scanner, one accurately determines the distance between the sample and the cantilever tip. Figure 2d plots the force as a function of traversed distance, rather

than of time, and is called the force-distance curve. 1 is the region of pull-in when the cantilever tip is attracted towards the surface until it meets the surface. 2 is the point at which the attractive force and the repulsive force cancel, where we extract the topography of our film; this is called contact point imaging (CPI) in the Quantitative Imaging (QI) mode. Beyond this is the regime of repulsion, denoted as 3, where the cantilever is pressed against the surface. 4 is the point of maximum indentation (δ), determined by the chosen setpoint of the measurement, where the tip deforms the surface in the normal direction. As the cantilever tip is retracted from the surface, the force between the two follows the red line shown. When the cantilever tip is pulled away from the surface, it shows greater attraction to the surface due to surface adhesion, marked in Fig. 2d as 5. From here, one infers the adhesive properties of the surface.

Although several nanomechanical properties of the sample surface can be extracted from force-distance curves, we focus here on the stiffness of the surface (see Supplementary Information Sections 1, 2, 3, 4 and 5). This parameter is read from the force-distance curve as the slope of region 3, highlighted in Fig. 2d. The corresponding slopes are then mapped and compared across all the samples under study. Nanoscale stiffness represents a better point of comparison than Young's modulus, since it is a directly measured quantity. The Young's modulus is extracted based on the measured stiffness using contact models and the Poisson ratio. Figure 2e shows the configuration of measuring nanomechanics on the semiconductors in this study. DNTT is shown on the very left. In the case of DNTT, the monoclinic unit cell is compact but swells in volume, vertically, along the z-axis direction upon sidechain substitution, i.e., in going from left to right in Fig. 2e to Ph-DNTT-Ph and C8-DNTT-C8. In RR-C8*-DNTT-C8* and SS-C8*-DNTT-C8*, the chiral side chains cause a slight expansion in the xy-plane. Figure 2f shows the herringbone packing of DNTT viewed from the top of the film, i.e., down the z-axis direction. This is the direction probed by the cantilever of the AFM and is related to the stiffness measured along the long axis of the conjugated molecular backbone. Within transistor devices, the charge-carrier mobility is measured in the xy-plane, shown in Fig. 2f, corresponding to the π-stacking direction within the herringbone structure. In this in-plane charge transport direction, ideas from deformation potential theory were previously used to connect transport with the modulus[28].

Force-distance curves were measured at every pixel within a selected region of the samples to map stiffness spatially. We choose to do measurements on areas of 5 × 5 square microns, 1 × 1 square microns, and 300 × 300 square nanometres, with 512 × 512 pixels per scan for the larger areas, thus measuring the stiffness with a spatial resolution of ca. 10 nm. Using cantilevers with tip radii of about 20 nm, calibrated individually before acquiring each scanned image (see Supplementary Information Section 2), we obtain complete stiffness information of the materials from our scans. For smaller areas of 300 nm x 300 nm, we reduced the number of data points per scan. Higher resolution here makes the scanning step smaller than the radius of the AFM probe, and only increases the duration of the scan without offering any new insight.

### Important subtleties in the nanomechanical measurements

The difficulty in performing nanomechanical characterisation of molecular surfaces lies in the fact that they are contact- and indentation-based, i.e., a surface must be 'felt' to probe its stiffness. Precise measurements of nanomechanical properties of molecular surfaces remain a fundamental challenge. Measurements rely on extremely fine contact and indentation, the signals of which are detected through changes in the position-sensitive photodiode (PSPD) response to AFM laser reflection. In other words, evaluating surface stiffness requires probing the surface in the intended direction, yet this process itself can perturb the mechanical properties of the target surface.

As described in Supplementary Information Section 5, we considered factors beyond typical plastic deformation or densification

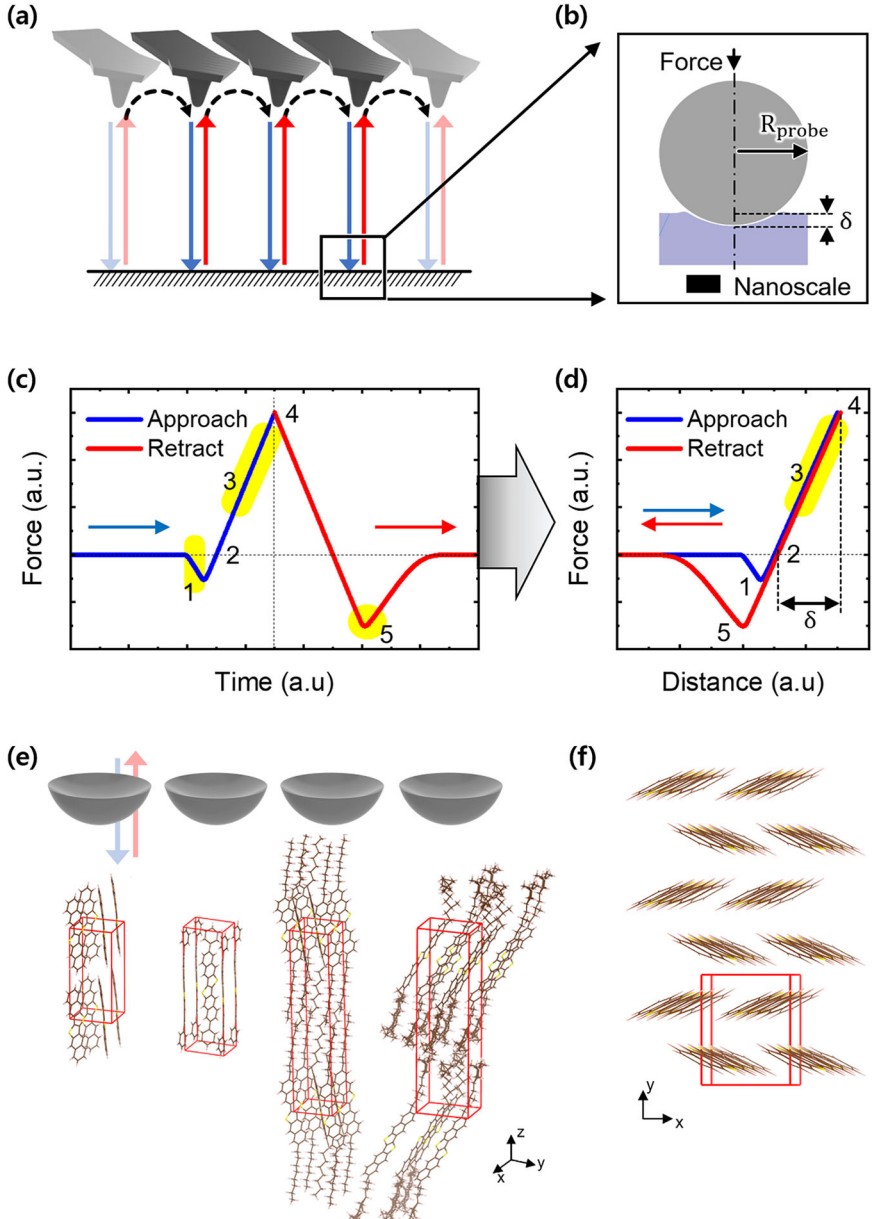

**Fig. 2 | Measurement of nanomechanics on an atomic force microscope. a** Point-to-point mode of measuring stiffness by quantifying force-distance curves between the tip of an AFM cantilever and a film surface. **b** Zoomed-in schematic view of the contact mechanics of an AFM tip with the film surface upon nanoscale elastic indentation. **c** Force vs time curve reflecting surface and mechanical interactions between the AFM tip and the thin-film surface during the approach and retraction of the AFM probe. **d** Force vs distance curves (force curves for short) calculated from the data in (**c**). **e** A representation of the mechanical measurement configuration, where the AFM probe lands on sidechains extending into the z-axis direction. **f** Herringbone patterns of DNTT molecules seen looking down the z-axis direction of the film. $\pi$-stacking is in the plane perpendicular to the direction of mechanical measurements using an AFM.

from indentation that could influence measurement outcomes. In particular, the potential lateral movement of the AFM tip and its impact on signal distortion were investigated.

Figure 3a–d present measurements on DNTT films, illustrating the importance of selecting an appropriate applied force. In Fig. 3a, forces exceeding 50 nN resulted in pronounced hysteresis between forward and retract curves, producing inconsistent and unreliable Young's modulus values. Consequently, measurements were performed with forces below 50 nN.

Figure 3b shows indentation depth and adhesion measurements on DNTT using a clean AFM tip. Reliable stiffness values were obtained with indentation depths of about 2 to 4 nm, corresponding approximately to a single molecular layer, using the maximum indentation from Supplementary Fig. 6. Generally, increasing the force setpoint

increased both indentation depth and adhesion, with adhesion ranging from 5 to 15 nN. Notably, at 50 and 70 nN, indentation depth plateaued and the expected correlation between adhesion and contact area $\pi Rd$, where $R$ is the tip radius and $d$ is the indentation depth, was disrupted.

Figure 3c and the 3D model inset illustrate tip contamination after repeated scans. Although the contamination layer was only around 30 to 40 nm thick, the measured Young's modulus was more than twice that of a clean surface. In Fig. 3d, the contamination resulted in up to a 30% reduction in indentation depth and a 30 to 100% increase in adhesion, indicating that contact occurred between organic aggregates rather than the original organic–inorganic interface.

Figure 3e shows adhesion fluctuations induced by tip contamination, suggesting that small molecules can detach during measurement, dynamically altering tip size and characteristics. These

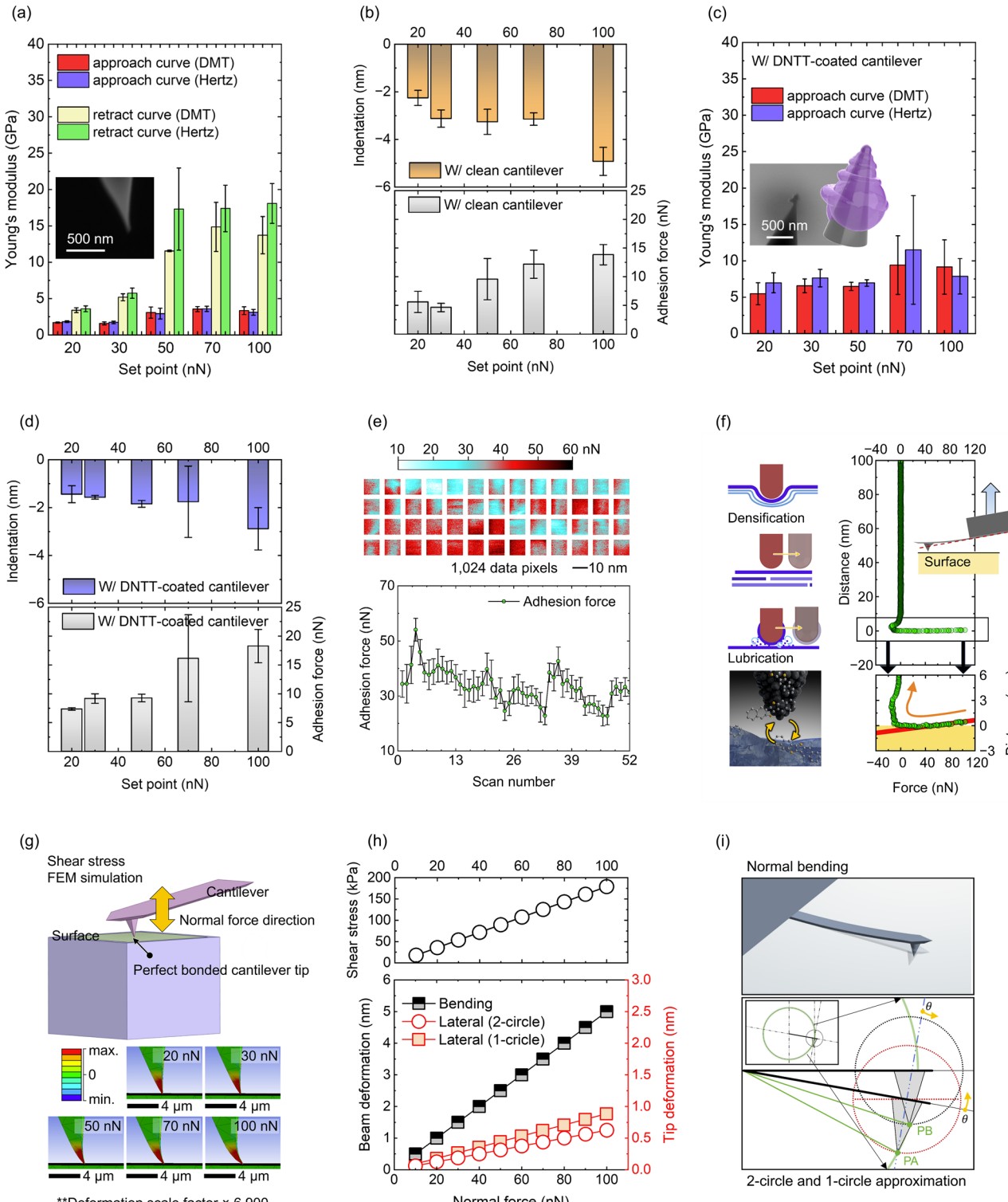

**Fig. 3 | Nanomechanical interactions between the AFM cantilever tip and DNTT molecular films. a–d** are specific to DNTT films. **a** Young's modulus extracted from forward and retract components of force–distance curves using a clean cantilever, and (**b**) the corresponding indentation depth and adhesion force are shown. **c, d** present measurements performed with a contaminated cantilever. **e** Spatially non-uniform adhesion highlights the effect of tip contamination, while (**f**) shows how tip contamination skews the force curves. **g** Finite Element Method (FEM) simulations illustrate AFM tip bending during surface contact and mechanical probing. **h** depicts lateral shear stress, cantilever beam deformation, and tip deformation during z-axis nanomechanical measurements. The upper graph is derived from the FEM simulations in (**g**), and the lower graph is based on the geometric model shown in (**i**). **i** shows the geometric model used to interpret lateral tip shifts.

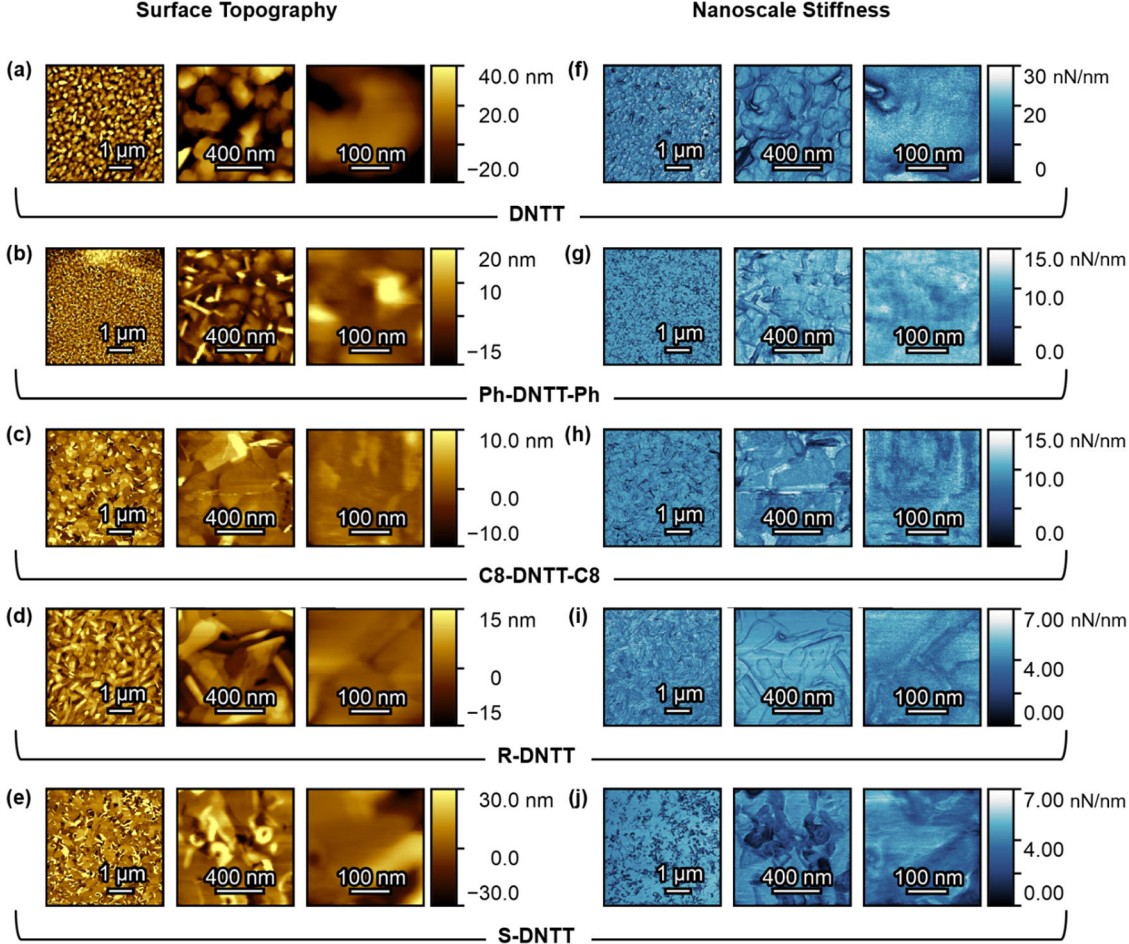

**Fig. 4 | Surface topography and nanoscale stiffness mapping in DNTT and its derivatives.** Surface topography of DNTT and its derivatives at three different length scales from the microscale to the nanoscale (left column in yellow).

Corresponding nanoscale stiffness of DNTT and its derivatives at three different length scales from the microscale to the nanoscale (right column in blue). Note the different scale bars.

adhesion variations imply the presence of changing amounts of contaminant material between the tip and the sample, potentially enabling molecular-level lubrication under in-plane tip movement.

Figure 3f illustrates negative slopes in force-distance curves arising from tip contamination, which could be misinterpreted as "negative stiffness." This may occur either due to actual tip motion causing a downward displacement despite decreasing load, or due to a lateral tip displacement during approach being misread as vertical bending during retraction. Both cases reflect abnormal tip behaviour and should be avoided for accurate nanomechanical assessment.

The plateau observed in Fig. 3a–d at high setpoints might appear attributable to densification under increased load. However, the negative slope during retraction in Fig. 3f cannot be explained solely by densification, as it is implausible for the tip to spontaneously move downward when the applied load decreases. Therefore, these distortions likely result from lateral tip movement or cantilever bending, which are interpreted as vertical movement of the cantilever.

To evaluate the lateral movement of the cantilever tip, additional FEM simulations were conducted (Fig. 3g–i). For a fully clamped cantilever, in-plane shear stress increased linearly with setpoint, and bending of the tip apex progressively increased from setpoints of 20 to 100 nN, becoming pronounced above 50 nN. In practice, weak adhesion (about 5 at 15 nN) is insufficient to maintain large bending, increasing the likelihood of lateral tip slip, which will alter the vertically measured signal on the PSPD.

Figure 3h, i further assess the impact of lateral movement on these vertical signals. Using two simplified models, (i) modelling the

cantilever and tip as rotating circles to simulate planar deformation, and (ii) representing the tip as a 2D arc, calculations showed that a lateral displacement of only 0.5 to 1 nm could be interpreted as vertical bending up to approximately 5 nm. For cantilevers with typical spring constants of 20 to 40 N/m, such bending would normally require 100 to 200 nN of applied load. Therefore, even minimal lateral movement can produce significant vertical signal artefacts.

In conclusion, contamination of the cantilever tip by detached DNTT molecules can severely distort measurement results. Contaminated tips reduce local contact pressure, potentially overestimating the Young's modulus. Dynamic changes in tip morphology and size make post hoc SEM evaluation of these effects difficult. Notably, lateral tip movement may be misinterpreted as vertical displacement, producing artificial signals.

Overall, Fig. 3 emphasises that careful consideration of tip-surface interactions is essential in AFM-based nanomechanical measurements. Supplementary Information Section 5 covers these topics in more detail. Understanding these interactions is critical for accurately determining intrinsic stiffness trends of DNTT and derivatives, and for ensuring reproducibility and consistency across samples.

### Comparison of nanomechanical properties for functionalised DNTT compounds

Figure 4 shows the measured surface topography and nanoscale stiffness of all five DNTT-based samples under investigation. Each of the rows in Fig. 4 show both the topography and stiffness at three different locations of the film, from the microscale to the nanoscale, as

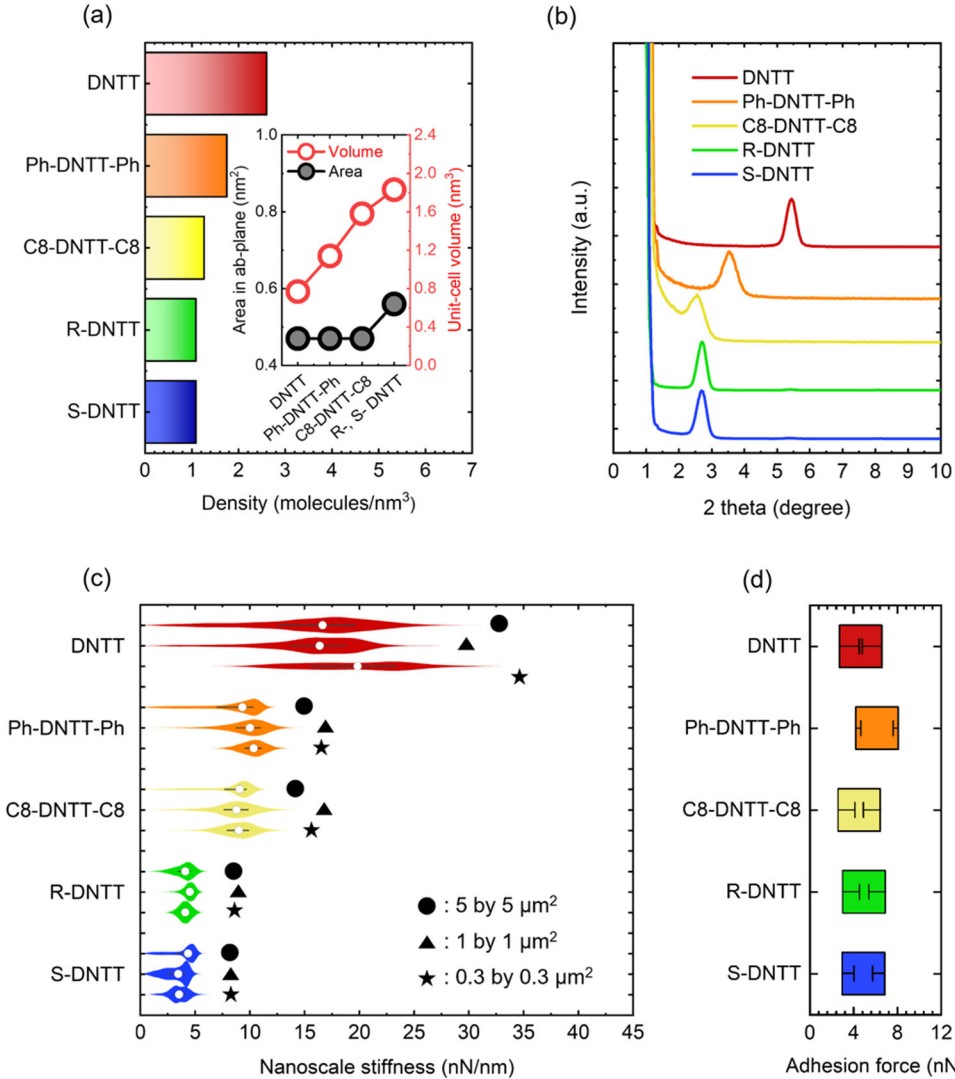

**Fig. 5 | Molecular density, X-ray diffraction, nanoscale stiffness, and surface adhesion of DNTT and its derivatives. a** Density of molecules per cubic nanometre in DNTT and its derivatives. The inset is a graph depicting volume changes based on variations in the unit cell *ab*- (xy) plane, and changes in length along the c-axis direction corresponding to the length of the sidechain. **b** Out-of-plane structural characterisation using X-ray diffraction in thin films of DNTT and its derivatives. As side chain substitution increases in length, going from DNTT to Ph-DNTT-Ph to C8-DNTT-C8, the out-of-plane peak in X-ray diffraction shifts to the left as shown, indicative of an expansion in the z-direction. In going from C8-DNTT-C8

to RR-C8*-DNTT-C8* and SS-C8*-DNTT-C8*, the volumetric expansion is due to changes in the *ab*- (xy) plane with little change in the out-of-plane direction. **c** Violin plots of nanoscale stiffness in DNTT and its derivatives measured over both microscale and nanoscale areas. The circle, triangle, and star, refer to three different areal scan sizes. **d** Compact plot of surface adhesion of DNTT and its derivatives. Adhesion does not change, indicating that the measured stiffness trend in (**c**) is intrinsic to the sample series and is not skewed by sample adhesion during measurement.

indicated by the scale bars. The stiffness of the family of films is investigated and compared in more detail in Fig. 5.

In Supplementary Information Section 3, we demonstrate that there is no strong correlation between the stiffness and the height topography in any of the DNTT derivatives. The fact that there is no clear correlation between stiffness and topographical roughness implies that the measurements are independent of each other and do not incorporate measurement artefacts from the way the tip scans across the sample during measurement. This is important to demonstrate, when assessing the reliability of nanoscale stiffness measurements. Often this can also be done by comparing line scans across topography and stiffness maps (see Supplementary Information Section 3) and ensuring that the features are not identical (which, for example, may be the case in samples with deep crevices where the AFM tip feels the texture around it more than it does underneath it).

As illustrated previously in Fig. 2e, alkylation on the DNTT molecule along the z-direction causes expansion in that direction. This leads to a reduction in molecular density, measured in molecules per cubic nanometre, as shown in Fig. 5a. The molecular density shown was calculated from the parameters of the unit cells of each material, tabulated in Supplementary Information Section 4. Along the vertical z-direction in which the stiffness is measured, a unit cell of DNTT contains only stiff cores, while the unit cell of C8-DNTT-C8 contains stiff cores with flexible alkyl sidechains. The unit cell of Ph-DNTT-Ph sits intermediate between the two, owing to its shorter phenyl groups. In other words, when longer sidechains are added onto the stiff DNTT core molecule, the unit cell expands in the vertical z-direction, the flexible sidechains begin to occupy more volume within the unit cell, and the molecular density reduces. It must be reiterated that the unit cells are monoclinic with vertical c-axes being nearly perpendicular to

the substrate's xy-plane, and so very close to the substrate's z-axis. The substrate's xy-plane is parallel to the *ab*-plane of the monoclinic unit cell. RR-C8*-DNTT-C8* and SS-C8*-DNTT-C8*, having two extra methyl groups in C8-sidechains, expand slightly into the *ab*-plane (see Supplementary Information Section 4), causing an ever so slight volumetric expansion beyond that of C8-DNTT-C8.

The unit cell expansion along the z-direction upon alkylation was also observed through out-of-plane X-ray diffraction measurements summarised in Fig. 5b. Although synchrotron-based measurements of structure are customary in the characterisation of organic systems for their high brilliance and angular beam spread, lab-based X-ray diffraction measurements offer insights with rapid turnaround and proved useful when studying the structural properties of DNTT[19,29]. From Fig. 5b, the peak originating from out-of-plane stacking is seen to shift to the left when going from DNTT to Ph-DNTT-Ph, and even further for C8-DNTT-C8. This is confirmation that the unit cell expands in the vertical direction upon side chain substitution (seeing as the molecules stand edge-on the substrate with side chains directed nearly along the z-axis perpendicular to the substrate)[30]. The unit cells of RR-C8*-DNTT-C8*and SS-C8*-DNTT-C8*, based on chiral C8-sidechains, slightly contract in the vertical direction with respect to C8-DNTT-C8, as confirmed in Fig. 5b.

Figure 5c shows violin plots of stiffness measured in nN/nm, appropriately highlighting the scale at which measurements are being done. There are three violin plots per molecular structure, reflecting three different resolutions at which the measurements were done (see Fig. 4) namely, $5\,\mu m \times 5\,\mu m$, $1\,\mu m \times 1\,\mu m$, and 300 nm × 300 nm. In Fig. 5c, there is agreement between the average values of stiffness measured at each length scale confirming reproducibility across length scales. The contribution from domain boundaries does not skew the average values. From Fig. 5c, the stiffness of DNTT is larger than the stiffness of Ph-DNTT-Ph, which is in turn larger than the stiffness of C8-DNTT-C8. RR-C8*-DNTT-C8* and SS-C8*-DNTT-C8*, namely the chiral versions of C8-DNTT-C8, are shown to be the least stiff and are comparable with each other.

The underlying reason why substituted DNTT derivatives are softer when probed along the vertical direction is because the incorporated flexible sidechains take up part of the volume in the unit cell. The π-conjugated core is hard, owing to its double bonds, while the alkyl side chains are soft. The combination of the two within a C8-DNTT-C8 unit cell renders it softer than DNTT. The stiffness trend in Fig. 5c is consonant with the changes in crystal density seen in Fig. 5a.

Figure 5d shows the measured surface adhesion forces on each film. The surface adhesion forces are understood to represent how sticky the samples are on the nanoscale. For measurements on stable films of the DNTT-family, the adhesion forces were comparable as shown in Fig. 5d. Had this not been the case, the stiffness values would not have been consistent or reliable because different surface adhesion can skew the portion of the force curves from which stiffness is extracted. Consistent adhesion values across all samples ensured reliability in the stiffness trend reported.

## Numerical calculations of mechanical properties in DNTT and C8-DNTT-C8

To understand the trend in our measurements of stiffness shown in Fig. 5c, we undertook numerical calculations of the Young's modulus. We identify the molecular films of DNTT and C8-DNTT-C8 as representing the most critical cases and calculate the modulus along all three orthogonal axes within their films. With these calculations, we can compare our out-of-plane AFM-based nanomechanical measurements with the calculated modulus along the z-axis.

Density-functional theory (DFT) calculations of the Young's modulus were carried out for DNTT and C8-DNTT-C8 based on the theory of elasticity[31]. These calculations used the FHI-aims software[32–34], and employed the B86bPBE functional[35,36], and the

XDM correction for describing van der Waals interactions[37,38]. First, the experimental crystal structures were fully optimised, relaxing both the atom positions and unit-cell parameters. A series of strained unit cells[39] were then generated from the optimised cells, and the corresponding stresses were computed (see Supplementary Information Section 6)[40]. To complement DFT calculations by including thermal and disorder effects, we also performed classical all-atom Molecular Dynamics (MD) simulations of the mechanical properties on DNTT and C8-DNTT-C8 using a modified Generalised Amber Force Field (GAFF)[41,42] (see Supplementary Information Section 7 for details on the parametrization procedure). Starting from DNTT[43] and C8-DNTT-C8[44] unit cells, two supercells containing 62 molecules each were created. We computed the stress tensor $\boldsymbol{\sigma}$ for different uniaxial strains ($\varepsilon$, up to ±1.0% with a 0.5% interval) and obtained the stiffness tensor $\boldsymbol{C}$ from Hooke's law. In both DFT calculations and MD simulations, from the stiffness tensor, we deduced the Young's modulus and the Poisson's ratio along any desired direction, by least squares fit of the $\boldsymbol{\sigma} = \boldsymbol{C} \cdot \boldsymbol{\varepsilon}$ relationship. Further computational details can be found in the Supplementary Information Sections 6 (DFT) and 7 (MD).

From the MD simulations, we observed for both DNTT and C8-DNTT-C8 that during contraction or expansion of the x-axis, the intermolecular distances $P_a$, $T_1$, and $T_2$, defined in Fig. 6a and plotted in Fig. 6d–f, increase (decrease) upon expansion (compression) of the unit cell. On the contrary, the distances $P_b$ and $P_c$, (see Fig. 6a, b), being perpendicular to the strain direction, are not affected by the deformation along x and are therefore not shown (see Supplementary Fig. 41). The same analysis was performed along y and z with similar conclusions, namely that the intermolecular distances perpendicular to the strain direction are not affected by it, while the evolution of the other characteristic distances follow the increase (decrease) of the strain magnitude, as displayed in Fig. 6c, g, h, i.

The Young's moduli computed at the DFT and MD levels are given in Table 1 and the values obtained with both methodologies are in broad agreement with each other. For the MD simulations, the standard deviations we computed for the $E_X$, $E_Y$ and $E_Z$ components are lower than 1%. The small differences in the xy-plane for C8-DNTT-C8 stand out; however, variations of this magnitude can be registered also by changing the functional or the force field parameters. Anisotropy of the elasticity is more robust to potential energy function changes[45]. It is worth noting that MD simulations are carried out at 300 K while DFT calculations are performed at 0 K: the inclusion of the kinetic energy in the MD simulations might permit exploring configurations, particularly for the flexible side chains in the xy-plane, that make the materials softer, and then explain the lower values obtained with MD for C8-DNTT-C8.

In comparing the mechanical properties of the two materials, clearly the DNTT crystal exhibits a larger resistance to deformations compared to C8-DNTT-C8 along the z-direction, consistent with the larger stiffness measured for the former system (see Fig. 5c). The larger $E_z$ for DNTT is in line with what was observed in previous calculations carried out with bare and alkylated benzothieno[3,2-b]-benzothiophene (BTBT)[46] and is attributed to the flexibility of C8 alkyl chains. The larger values of $E_z$ with respect to $E_X$ and $E_Y$ in both systems are explained by considering the parallel orientation of the molecular long axis with respect to the cell z-axis.

Table 1 also shows that, unlike $E_z$, the moduli along the x- and y-directions increase when going from DNTT to C8-DNTT-C8, although to a much greater extent with DFT than with MD. In both systems the elastic responses in the xy-plane show nearly identical values, i.e., $E_x \sim E_y$, not surprisingly in view of the similar nearest-neighbour molecular interactions within the *ab*- (xy) plane (see Fig. 2f). The similarity of the in-plane Young's moduli is substantiated by observing that the slopes of characteristic distances as a function of the x- and y-strain, obtained with MD simulations, are similar overall (see Fig. 6). Instead, the significant difference in $E_z$ is accompanied by a larger variation of $P_C$ versus $\varepsilon_z$ for C8-DNTT-C8 as compared to DNTT (see Fig. 6c).

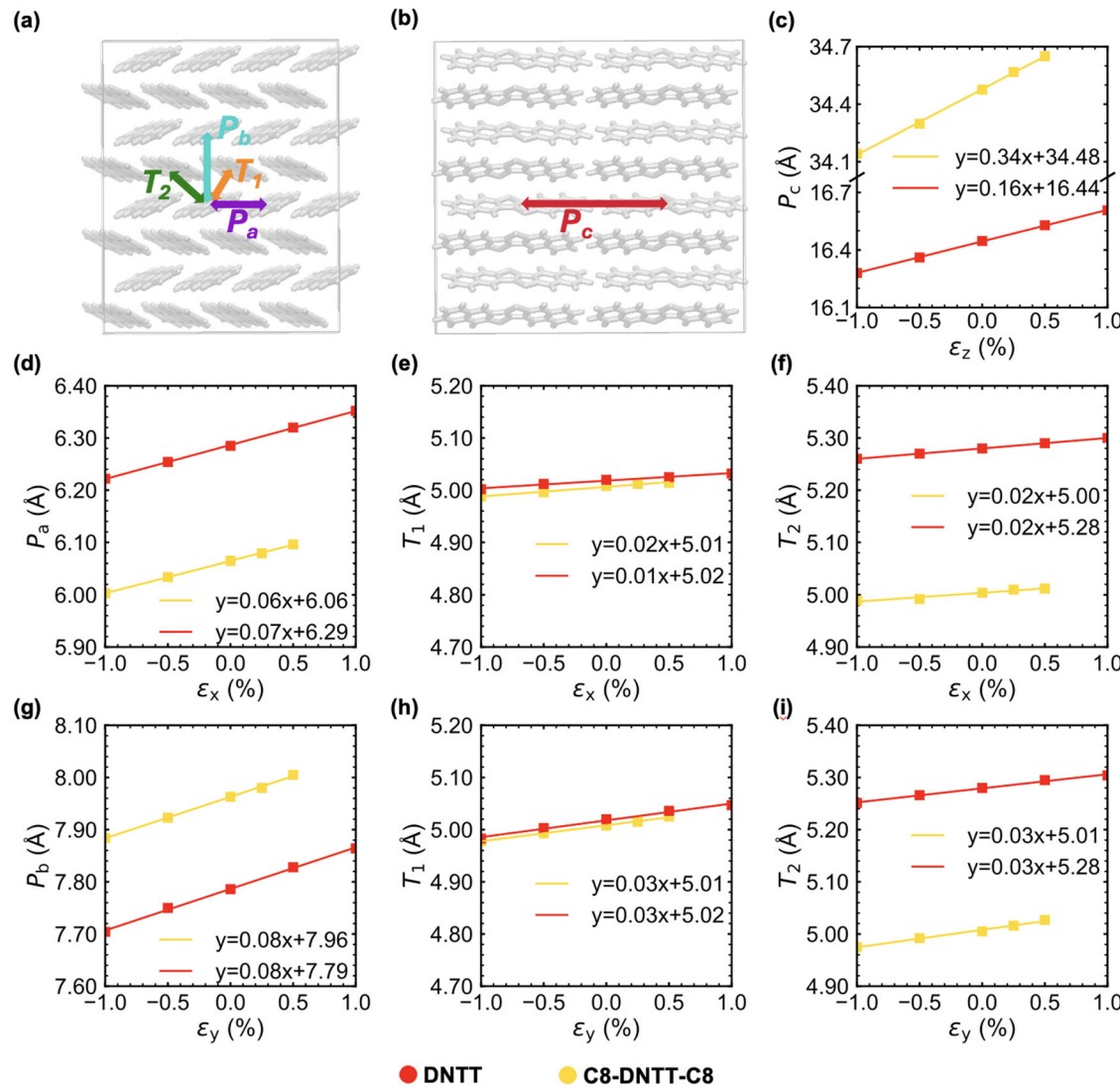

**Fig. 6 | Comparison of the structure change in the herringbone pattern for DNTT (in red) and C8-DNTT-C8 (in yellow) as obtained from molecular dynamics simulations.** **a** shows the supercell in the xy-plane: the characteristic structural distances, namely $P_a$, $P_b$, $T_1$, and $T_2$, are represented by the purple, blue, orange, and green arrows, respectively. **b** shows the supercell in the yz-plane highlighting the distance $P_c$ (red arrow). **c** shows the time-averaged distance $P_c$ as a function of strain $\varepsilon_z$ along z. **d**–**f** show the time-averaged distances $P_a$, $T_1$, and $T_2$ as a function of strain $\varepsilon_x$ along x, and (**g**–**i**) show the time-averaged distances $P_b$, $T_1$, and $T_2$ as a function of strain $\varepsilon_y$ along y. Straight lines and equations are the results of least-square fits. The units of the slopes and the intercepts are in are in Å/% and Å, respectively.

**Table 1 | Computed Young's moduli in GPa of DNTT and C8-DNTT-C8 along the Cartesian axes using DFT and MD**

|  | DNTT | | C8-DNTT-C8 | |
|---|---|---|---|---|
|  | DFT | MD | DFT | MD |
| $E_x$, (100) | 3.3 | 2.2 | 5.2 | 2.5 |
| $E_y$, (010) | 3.3 | 2.0 | 5.8 | 2.2 |
| $E_z$, (001) | 18.3 | 21.7 | 14.5 | 13.3 |

Overall, our MD and DFT results predict a large anisotropy in the Young's modulus and report an $E_z$ agreeing with experimentally measured trends in the stiffness.

## Discussion

### Link between mechanical properties and charge-carrier mobility

In prior work on non-polar inorganic semiconductors, such as silicon, acoustic deformation potential theory was used to link their electronic properties with their mechanical properties[11,47], The theory bases itself on the assumption that charge-carrier scattering is dominated by acoustic phonons and that there are no interfacial grain boundaries or chemical impurities that cause scattering.

In more recent studies on DNTT and C10-DNTT, acoustic deformation potential theory, tailored for transport in two dimensions, linked the in-plane charge-carrier mobility with the elastic modulus. The relationship between these quantities was directly proportional according to this theory[28]. The theory also predicted an inverse proportionality of the mobility on the acoustic deformation potential constant, which accounts for electron-phonon coupling, and a direct proportionality to the degree of carrier confinement in the direction perpendicular to the plane of transport. Because this theory was developed to understand the relationship between the modulus and the mobility in the plane of transport, it cannot be applied to the vertical z-direction in DNTT and alkylated DNTT due to a lack of $\pi$-stacking in that direction. Broadly, acoustic deformation potential theory suggests that stiffer lattices along the direction of charge transport are responsible for larger charge-carrier mobilities.

The two-dimensional acoustic deformation potential theory deployed in Northrup's published work on C10-DNTT used values for the elastic modulus in the tens of GPa[28]. The values correspond to simultaneous deformation along the *a* and *b* axes of the monoclinic unit cell in the 2D lattice plane where hole conduction takes place, and perpendicular to the C10 alkylated chains. Values in the tens of GPa for the elastic constant is typical for high mobility organic single crystals such as rubrene[48–50] In C10-DNTT, a modulus of tens of GPa corresponds to calculated mobilities between 50 and 100 cm²/Vs[28]. Such large values have never been experimentally reported. Most reported mobilities hover around 10 cm²/Vs[51], likely because of extrinsic experimental factors such as grain boundaries, contact resistance, and so forth. From our own measurements of field effect transistors fabricated from organic molecules studied here (see Supplementary Information Section 8), the charge-carrier mobilities are as follows: DNTT has a mobility of 2.3 +/− 0.1 cm²/Vs[52], Ph-DNTT-Ph has a mobility of 3.7 +/− 0.3 cm²/Vs[53], C8-DNTT-C8 has a mobility of 4.8 +/− 0.3 cm²/Vs[52], RR-C8*-DNTT-C8* has a mobility of 0.51 +/− 0.08 cm²/Vs[19], and SS-C8*-DNTT-C8* has a mobility of 0.59 +/− 0.01 cm²/Vs[19]. Our dataset, shown in Supplementary Information Section 8 for DNTT, C8-DNTT-C8, and its chiral versions is based on mobilities extracted from field effect transistors fabricated on AlO$_x$ coated with a self-assembled monolayer (see Methods Section). All the organic transistors, including those fabricated using Ph-DNTT-Ph[54], used a fully consistent processing protocol and device architecture. Similar charge-carrier mobility values measured in organic transistors fabricated from DNTT and its derivatives have been published in earlier reports[13–19,43,44,51,55–66]. These charge-carrier mobilities are measured in the *ab*-plane of the monoclinic unit cell, along the direction of $\pi$-stacking, and not along the direction of the alkyl side chains that nanomechanics probes.

The link between stiffness and electron-phonon coupling is not straightforward because the evolution of the transfer integrals with intermolecular distances is highly non-linear. The differences in charge-carrier mobility are neither solely attributed to differences in stiffness nor solely to differences in electron-phonon coupling[67]. So far, the literature has mainly considered differences in electron-phonon couplings. For example, a recent report suggested that the difference in mobility between the two DNTT derivatives may arise from a positive product of the electronic couplings $T_2 \cdot P_a \cdot T_1 > 0$ in C8-DNTT-C8, compared to a negative product in DNTT. It was also suggested that the electronic couplings in C8-DNTT-C8 are more isotropic compared to DNTT, allowing for better charge delocalisation in C8-DNTT-C8 compared to DNTT[67]. Our work suggests that stiffness is also a factor to be considered. A stiffer material in the direction of transport, we speculate, may limit the amplitude of intermolecular vibrations modulating the intermolecular distance between the molecules.

From a mobility and modulus standpoint, C8-DNTT-C8 has a larger in-plane charge-carrier mobility and a larger calculated in-plane modulus compared to DNTT (see Table 1 and Supplementary Section 8). In other words, a stiffer lattice in the direction of fast charge transport does seem to correspond to higher charge-carrier mobility. In future, if the direction of charge transport is accessible within an atomic force microscope, careful and consistent nanomechanical measurements such as the ones shown in this work may, more generally, address the question of whether it is necessary to make crystalline organic semiconductors stiffer on the nanoscale to realise high mobility.

## Conclusion

The mechanical stiffness of a molecular semiconductor depends on solid-state intermolecular interactions, as well as on single-molecule properties. In this work, we demonstrated how consistent and careful AFM-based nanomechanical measurements allow one to quantify minute changes to the stiffness of a family of organic semiconductor based on DNTT, where the aromatic core is systematically substituted by soft, flexible alkyl side chains. Thus, while alkylation is expected to result in reduced stiffness, our work presents the first direct measurement of the impact of alkylation on mechanical properties. Calculations based on molecular dynamics and density-functional theory confirm the experimental trend that increased alkylation reduces stiffness in the out-of-plane direction, normal to the surface. The ability to perform reliable, artefact-free, AFM-based nanomechanical measurements of molecular lattices is a key step towards assessing proposed connections between the stiffness and charge-carrier mobility of organic semiconductors.

## Methods
### Materials

DNTT was purchased from Sigma-Aldrich and used as received. C8-DNTT-C8 and Ph-DNTT-Ph were supplied by Luminescence Technology Corp. (Lumtec) and Nippon Kayaku. RR-C8*-DNTT-C8* and SS-C8*-DNTT-C8* were synthesised according to previously described procedures[19].

### Thin film deposition

The organic semiconductor thin films were fabricated on highly doped silicon wafers with 300 nm of SiO₂ (Christian-Albrecht University of Kiel, Institute for Electrical Engineering and Information Technology). Organic semiconductors were evaporated in vacuum (UNIVEX 300, Leybold GmbH, pressure of ~ 10⁻⁶ mbar, deposition rate of 0.3 Å s⁻¹ for DNTT and of 0.1 Å s⁻¹ for Ph-DNTT-Ph, C8-DNTT-C8, RR-C8*-DNTT-C8* and SS-C8*-DNTT-C8*, nominal thickness of ca. 50 nm, monitored by a quartz crystal microbalance) onto the substrates which were held at temperature of 80 °C for DNTT, 90 °C for Ph-DNTT-Ph, 100 °C for C8-DNTT-C8, and at 40 °C for RR-C8*-DNTT-C8* and SS-C8*-DNTT-C8*. Immediately following the deposition of OSCs onto the substrates, the samples were promptly sealed in vacuum packs at room temperature. They remained shielded from ambient conditions and light throughout the duration between deposition and nanomechanical measurements.

### Nano-mechanical measurements

All AFM measurements were performed with the NanoWizard®3 AFM (Bruker, USA) in the ambient at 23 °C. Nano-mechanical measurements using Quantitative Imaging (QI) were conducted on all samples with Tap300DLC cantilevers (Budget Sensors, Bulgaria) in areas of 5 by 5, 1 by 1, and 0.3 by 0.3 μm². Additional measurements were done on a Park Systems NX10 AFM. To avoid contamination, a new cantilever was used for each sample, and the normal bending spring constant and sensitivity was determined for each cantilever before the measurements. The normal bending spring constant using the Thermal Tune method was approximately 24 N/m. QI map analysis was performed using the JPK Data Processing software (v6.1.120), and Gwyddion (v2.64).

### XRD measurements

Grazing Incidence X-ray Diffraction (GIXRD) analyses, both in-plane and out-of-plane, were conducted using a multi-purpose X-ray diffractometer (SmartLab, Rigaku, Japan) using Cu Kα (λ = 1.5406 Å) radiation generated by an X-ray generator at 45 kV and 200 mA. We show only the out-of-plane data in the work since it is relevant to our discussion.

### Field effect transistor fabrication and electrical measurements

Field effect transistors with DNTT, C8-DNTT-C8 and chiral DNTTs were fabricated on highly doped silicon wafers coated with 30 nm of Al₂O₃ (Christian-Albrecht University of Kiel, Institute for Electrical Engineering and Information Technology). The Al₂O₃ substrates underwent oxygen plasma treatment (Diener Electronic; oxygen flow rate 20 sccm, pressure 0.50 mbar, plasma power 100 W, duration 2 min) and then immersed overnight in 1.5 mM solution of n-tetradecylphosphonic acid (TDPA, Sigma-Aldrich) in 2-propanol

(Acros Organics) to obtain a 1.5 nm-thick self-assembled monolayer (SAM), resulting in an $Al_2O_3$/SAM dielectric possessing a capacitance ($C_i$) of 185.5 nF cm$^{-2}$. Subsequently, the substrates were rinsed first in 2-propanol then in deionized water and finally in 2-propanol again and dried on a hot plate at 100 °C for 10 min. To fabricate bottom-gate bottom-contact (BGBC) devices, gold contacts were deposited on the gate-dielectric substrates at room temperature via vacuum deposition (UNIVEX 300, Leybold GmbH; pressure of 10$^{-5}$ mbar, deposition rate of 0.5 Å s$^{-1}$, nominal thickness of 50 nm, monitored using a quartz crystal microbalance). Afterward, the substrates were immersed in a 10 mM pentafluoro benzenethiol (PFBT, Alfa Aesar) solution in 2-propanol for 30 minutes to form a SAM on the gold bottom contacts. The substrates were then rinsed with 2-propanol and dried. OSC was deposited onto the substrates by vacuum evaporation (pressure of 10$^{-6}$ mbar, deposition rate of 0.3 Å s$^{-1}$ for DNTT and 0.1 Å s$^{-1}$ for C8-DNTT-C8, RR-C8*-DNTT-C8* and SS-C8*-DNTT-C8* nominal thickness of 25 nm, monitored by a quartz crystal microbalance and confirmed by AFM measurements) using a shadow mask, with the substrates maintained at specific temperatures (70 °C for DNTT, 100 °C for C8-DNTT-C8 and 40 °C for RR-C8*-DNTT-C8* and SS-C8*-DNTT-C8*). This process yielded the final devices with a channel width ($W$) of 480 μm and a channel length ($L$) of 215 μm.

Electrical measurements were performed at room temperature in ambient air using an Agilent 4155 C Semiconductor Parameter Analyser. The transfer characteristics were measured in the linear regime at a drain-source voltage ($V_d$) of − 0.1 V. Field-effect mobility was calculated using the conventional gradual channel approximation model with equations applicable to the linear regime:

$$\mu = \frac{L}{WC_iV_d}\frac{\partial I_d}{\partial V_g}\bigg|_{V_d}$$

where $I_d$ is the drain current and $V_g$ the gate voltage. The IV characterstics and mobilities are summarised in the Supplementary Information Section 8.

## Data availability

The data used in this study are presented in the text and the Supplementary Information. AFM data can be accessed at the online repository https://doi.org/10.17863/CAM.123709. Additional data may be available from the corresponding authors upon request.

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

## Acknowledgements

D. Venkateshvaran acknowledges the Royal Society for funding in the form of a Royal Society University Research Fellowship (Royal Society Reference No. URF/R1/201590). L.F. would like to acknowledge funding from BBSRC (UKRI) through a prosperity partnership grant (BB/Y003225/1). G. Schweicher is a Belgian National Fund for Scientific Research (FNRS) Research Associate. G. Schweicher acknowledges financial support from the Francqui Foundation (Francqui Start-Up Grant) and thanks the FNRS for financial support through research project COHERENCE2 (No. F.4536.23). D. Venkateshvaran and G. Schweicher acknowledge funding from the Wiener-Anspach Foundation (FWA). F. Modesti and M. Volpi acknowledge the financial support from the European Union's Horizon 2020 Research and Innovation Programme under the Marie Sklodowska-Curie Grant No 811284. F. Modesti acknowledges

discussions with Dr. Peter Erk. D. Venkateshvaran, and K.-H. Hwang thank Dr I. Dobryden for discussions on nanoscale mechanical measurements. E.R. Johnson is grateful for financial support from the Natural Sciences and Engineering Research Council (NSERC) of Canada, and from the Royal Society through a Wolfson Visiting Fellowship. E.R. Johnson and C.J. Nickerson thank the Atlantic Computing Excellence Network (ACENET) for computational resources. C.J. Nickerson acknowledges financial support from a Nova Scotia Graduate Scholarship. Yves Geerts thanks the Belgian National Fund for Scientific Research (FNRS) for financial support through research projects: Pi-Fast PDR T.0072.18, PICHIR PDR T.0094.22, DIFFRA GEQ U.G001.19, POLYP EQP U.N032.21 F, POLYP2 EQP U.N03323F, CHIRI CDR J. 0088.24, CISSCA WEAVE T.W.023.23, and CHISUB EOS no. 40007495. The authors thank Nippon Kayaku for supplying Ph-DNTT-Ph.

## Author contributions

K.H.H. performed the nanomechanical measurements on multiple AFMs in the groups of P.M.C and D.V. K.H.H. also performed the X-ray diffraction measurements. D.B. and S.C. carried out the numerical calculations of nanomechanical properties under the supervision of Y.O. and L.M. D.B. and S.C. contributed equally. C.J.N. performed the density-functional calculations of Young's modulus under the supervision of E.R.J. F.M. fabricated the films and characterised them together with G.S. F.M., and M.V. carried out electrical measurements of mobilities supervised by Y.G. and G.S. M.V., F.M., G.S., and Y.G. synthesised and characterised the chiral molecules. G.S. supervised the chemistry of growth and the electrical property measurements performed within the consortium. M.T.R.C. carried out the statistical analysis on nanomechanical data. D.V., P.M.C., and L.F. established laboratory facilities and supervised the nanomechanical property measurements within the consortium. M.G. offered insight into the interpretation of the measurements. L.J.S. performed the first round of measurements on the films and helped set up the project with D.V. K.H.H. prepared the figures for the paper with direction from D.V. D.V. and K.H.H. designed the project and wrote the paper together with input from all authors.

## Competing interests

The authors declare no competing interests.
