## [Transparent Peer Review file · Nature Communications]

Measuring the molecular origins of stiffness in organic semiconductors

Corresponding Author: Dr Deepak Venkateshvaran

A version of this paper was originally rejected for publication by Nature Communications, however that decision was reconsidered after appeal by the authors.

Version 1:

Reviewer comments:

Reviewer #1

(Remarks to the Author)

The article by Kwang and coworkers addresses an important but underexplored question, which is the relationship between the molecular structure of an organic semiconductor film and its nanomechanical properties, particularly its stiffness. The work is carefully done and explained and the manuscript is clearly written. However, with the current version of the article I find it hard to recognise what exactly its novel contribution is.

The title suggests that the article is addressing the relationship between molecular structure and stiffness. To some extent the paper addresses this question and shows that in crystalline molecular thin films that are oriented with the long molecular axis normal to the substrate, that the stiffness along this axis depends on the amount of non-conjugated material along that axis, with longer side chains leading to lower stiffness for a given conjugated unit. This makes sense given that the fused ring pi-conjugated part of the molecule is itself stiffer than the more flexible end groups and given that the selected materials are fairly crystalline (so the molecular orientation is more or less known). This result is clear, though not surprising.

The abstract suggests that the article concerns the relationship between stiffness and electronic (charge transport) properties. Here the relationship is not so clear. Stiffness is measured in the z direction, FET mobility is measured in the xy plane. The authors explain the higher mobility of C8-DNTT than DNTT in terms of a higher calculated stiffness in that plane, but this has no relationship to their careful measurements of stiffness in the orthogonal direction and moreover the difference in calculated stiffness in the xy plane between the two materials is not larger than the error on the theoretical method that the authors tell us to expect. As well, they show very clearly that the calculated stiffness in the z direction is significantly larger than any orthogonal direction and yet, charge carrier transport is by far the slowest in that (z) direction, leaving the conclusion that high stiffness can lead to high mobility rather undermined. Previous work rationalising the higher mobility of C8-DNTT than DNTT appealed to electron-phonon coupling yet the theoretical methods that the authors select here to explain their results do not, as far as I can see, account for electron-phonon coupling. The authors argue that electron-phonon coupling is not the only parameter that matter and here I am sure they are right, but I don't find it very clear from the paper in its current form what physical explanation they are offering us instead.

I like what the authors are trying to do but I think the authors need to clarify their story before I could recommend it for publication. One approach might be to improve the margins of error on the theoretical results by validation of the method on other material systems. It would be helpful if they could include the impact of vibrational modes in that analysis, or discuss them explicitly if they are already included in the work done. Another approach might be to find a way to probe transport and stiffness in the same directions. Or to focus the work primarily on structure-nanomechanical property relations, where the paper and the extensive Supporting information are very strong.

A final general concern is that all the analysis is done assuming that the molecular films are crystalline are oriented in a particular sense, however we don't know how crystalline they are. One might expect that any amorphous material and / or grain boundaries would affect both nanomechanical measurements and mobility. Can the authors say how crystalline the films are and can they quantify the relative degree of crystallinity of the different DNTT derivatives. Can they rule out the possibility that differences between materials could be caused by different degrees of crystallinity or orientation?

Apart from these general concerns I have some minor points:

The term 'pi-conjugation' is used several times in a way that suggests the authors mean pi stacking. I would understand that the direction of pi-conjugation is the long axis of the molecule, along which conjugation occurs, not an orthogonal direction.

On p.10 the authors stress that there is no correlation between nanoscale stiffness and topography yet the figure 3 seems to be showing that there is some correlation in the spatial features. If the apparent similarities between these images are due to some larger scale features like cracks then it would be helpful to remark on that when discussing the comparison.

p.11, please say briefly how density is determined. I am guessing this refer to a density worked out from the solved crystal structure and not a measurement that is relevant to the thin films.

P12, is it not trivial that molecular density reduces as side chains get longer?

The x-ray diffraction data on p.14 are presented as showing the crystallinity of the films – does that mean degree of crystallinity and if so how? They are also claimed to rule out polymorphism but that may be too strong, depending on the sensitivity of the measurement.

I would dispute that the DFT and MD are really in broad agreement with each other. Do you have good reasons to expect the methods to agree? For example did you use the same functionals that were selected for DFT to tune the force field used in MD?

The sentence on page 4 that tells the reader that they should have confidence in the work because the authors have plenty of experience with these materials read rather odd to me. We generally have to assume that authors are confident in the work they submit for publication.

Reviewer #2

(Remarks to the Author)

The paper investigates the stiffness and electronic properties of a family of high mobility organic semiconductors based on dinaphtho[2,3-b:2',3'-f]thieno[3,2-b]thiophene (DNTT) derivatives. The authors focus on understanding the interrelationship between molecular structure, nanoscale mechanical stiffness, and charge carrier mobility in organic semiconductor films.

One major concern I have with the paper is the relevance of the work – the authors measure the mechanical properties in the z-direction, find that it doesn't correlate well with the reported charge mobilities because charge mobility is measured in the xy-direction, thus calculate the expected mechanical property in the xy-direction and correlate that to the charge mobility. While they find that there is some correlation, they also ultimately conclude that the charge mobility isn't as simple as that and is also related to the electronic coupling between molecules. In some ways, these are all expected findings and it is unclear what new knowledge it provides to the field. To make the paper suitable for Nature Communications, I'm wondering if the paper should be more focused on the unique aspects of the work, not on the failed hypothesis/narrative.

In particular, what must have been an immense amount of work are the AFM measurements. The authors are to be applauded to the attention in detail for their measurements using AFM – it's a shame that much of the effort and detail has been sidelined to the SI because one can only imagine that it must have been a huge amount of effort to elucidate the details of this measurement. Similarly, it's a bit of a shame that the simulation is tucked away in the SI.

1. Figure 4b – the y-axis refers to the ab-plane but it might be helpful to refer to it in terms of xyz to be consistent with the discussion. I recognize that they are not quite the same thing but given that the authors themselves state that "unit cells are monoclinic with vertical c-axes being nearly perpendicular to the substrate's xy-plane, and so very close to the substrate's z-axis." I think it may be ok. How were the data for Figure 4b obtained?
2. Page 14: I found the following statement a bit confusing "This is an indication of the crystallinity of the films". How does the shift in peak position refer to the crystallinity of the film?
3. Page 16: I found the first paragraph to be very repetitive compared to what was stated on the previous page.
4. Can the authors comment on the chiral derivatives more? In line with the theme of the paper, how does the mechanical property reflect in the charge mobility in the two examples investigated?

Reviewer #3

(Remarks to the Author)

This work offers an insightful investigation into the origins of stiffness in organic semiconductor (OSC) layers by revealing lattice-dependent mechanical properties through atomic force microscopy (AFM). To achieve this, the authors developed an out-of-plane AFM characterization method tailored for probing mechanical properties. Notably, the results successfully disentangle the effects of crystalline structure on stiffness while elucidating the relationship between surface mechanical properties and carrier transport. This study provides a valuable strategy to guide researchers in exploring surface physical properties and informs molecular design for optimized elastic and electrical performance. There are additional comments which must be addressed for clearer explanation of this manuscript.

(1) As demonstrated in this manuscript, the molecular lattice plays a crucial role in determining the stiffness of OSC thin films. The author analyzed the crystalline unit cell and molecular stacking at the nanoscale (as shown in SI Section 4). However, how can the geometry of an individual molecule on the substrate be confirmed? Specifically, how can the molecular orientation at the AFM tip contact point be determined? Furthermore, does the molecular orientation influence the stiffness, and if so, how?

(2) Could the domain size, surface roughness, and film thickness influence this analysis, given that the shear stress in AFM measurements can be significantly impacted by the contact edge of the thin film?

(3) The author demonstrated how the intrinsic factor--molecular structure--influence the stiffness of semiconductor thin films. DNTT and its derivatives exhibit excellent crystalline film quality. Could this approach serve as a universal strategy for other small molecules, such as Naphthalene diimide (NDI) molecules with larger π -conjugated cores?

(4) It is intriguing that the author links the carrier transport properties with the mechanical properties, showing that stiffer lattices along the charge transport direction contribute to higher charge carrier mobilities. However, it is challenging to directly correlate intermolecular hopping with lattice stiffness. Does the stiffer lattice enhance pi-pi overlap and facilitate more efficient carrier transport pathways?

Version 2:

Reviewer comments:

Reviewer #2

(Remarks to the Author)

In this revised version, the authors have carefully addressed the concerns raised by the previous reviewers. The authors have demonstrated careful control of AFM parameters to extract accurate measurements for the mechanical properties and they have validated their results through support from DFT and MD. The technique, in general, would be applicable other organic semiconductors. I believe that the paper is now appropriate for publication.

Things for the authors to consider:

1. Given that the mechanical property measurement is in the z-direction, have the authors considered correlating that to the charge mobility in the z-direction by, for example, performing SCLC measurements? Alternatively, perhaps use the AFM to obtain conductivity measurements (c-AFM)?

2. I'm also wondering if there is anyway to extract two different values for the stiffness. One for the aromatic core and one for the alkyl region. I'm naively wondering if as the tip approaches the surface, one could first get the alkyl region value and then the stiff of the aromatic core as the tip approaches the surface more.

Reviewer #3

(Remarks to the Author)

The current manuscript, revised by Hwang, deepens the understanding of the nanomechanical properties influenced by the hierarchical structure, through artifact-free AFM technique. The comprehensive characterization and analysis strengthen the significance and reliability of the research, with the systematic analysis contributing to a better understanding of the relationship between stiffness and charge carrier mobility. The systematic analysis contributes to the understand the relationship between the stiffness and charge carrier mobility. There are still several issues that need to be addressed.

1. The authors performed GIXRD measurements on DNTT and its derivatives. Do they have any corresponding 2D diffraction patterns or additional data? In fact, the diffraction results provide a statistical analysis of crystalline stacking in regions exposed to X-ray irradiation. While MD and DFT simulations are included for comparison, the relationship between molecular stacking orientation and mechanical properties needs clearer explanation and further clarification.

2. The carrier transport properties in this study were assessed through FET devices. Have the authors considered analyzing the electrical performance/carrier transport via conductive-AFM in conjunction with the nanomechanical measurements? This could provide additional insight into the electrical behavior at the molecular level.

3. In addition, the language expression and structural framework of the article are not very reasonable, which creates difficulties in reading. The author should revise the entire manuscript, optimizing aspects such as formatting and paragraph organization.

Reviewer #4

(Remarks to the Author)

The authors present a joint experimental and theoretical investigation of the relationship between molecular structure, packing configuration, and stiffness of molecular crystals. Here, the molecules have a consistent pi-conjugated core structure, and differ by side chains appended symmetrically on the ends of the long axis of the core; the side chains range from hydrogen (unsubstituted), phenyl, and alkyl (with three variations, though the overall lengths are the same). Notably, the side chains do change the tilt of the molecular long axis with respect to the surface and direction of the mechanical force measurements.

The authors have done a commendable job in responding to the first round of reviewer comments. Indeed, the major focus of the manuscript was changed. However, the authors have not sufficiently addressed what the “noteworthy results” of the work are. That careful measurements have been done, and subtleties that need to be considered when making the measurements are noted, is important work; however, this is not necessarily noteworthy. The outcomes of the physical measurements, and likewise of the DFT and MD simulations that are validated by experiment, are not surprising given the natures of the molecules and substitutions explored. Hence, the overall significance of the work is limited. The data analyses are complete, and appear to be interpreted correctly and appropriately. The work does not appear to make spurious claims that go beyond what is measured / calculated. This is a solid paper, and the work is well done.

Version 3:

Reviewer comments:

Reviewer #2

(Remarks to the Author)

I believe that the authors have addressed the comments as much as they are able to at this time.

Reviewer #3

(Remarks to the Author)

The authors have made appropriate revisions. I recommend acceptance of the manuscript.

Response to Reviewers Comments

We begin by thanking the reviewers for taking time to look at the improvements made after the first round of review and for raising further points in need of clarification.

We also welcome the supportive comments of the new Reviewer # 4 who has contributed to the second round of review.

We've taken time to carefully address, and incorporate, the suggestions of the reviewers during the second review round. Doing so has improved the quality of our manuscript further, we believe.

Below, the reviewers' comments are shown in black, and ours in purple.

All relevant changes have been highlighted in yellow in the main manuscript.

Reviewer #2 (Remarks to the Author):

In this revised version, the authors have carefully addressed the concerns raised by the previous reviewers. The authors have demonstrated careful control of AFM parameters to extract accurate measurements for the mechanical properties and they have validated their results through support from DFT and MD. The technique, in general, would be applicable other organic semiconductors. I believe that the paper is now appropriate for publication.

Things for the authors to consider:

1. Given that the mechanical property measurement is in the z-direction, have the authors considered correlating that to the charge mobility in the z-direction by, for example, performing SCLC measurements? Alternatively, perhaps use the AFM to obtain conductivity measurements (c-AFM)?

Conductive-AFM (C-AFM) is indeed a relevant technique using which both mechanical properties and electronic properties can be consonantly measured along the z-direction. In C-AFM, as the reviewer suggested, SCLC measurements can be performed over the scanned area by measuring IV characteristics at every pixel. From these IV characteristics, the SCLC carrier mobility can be extracted at each pixel in the same way we've extracted the stiffness.

The suggested combination of SCLC/C-AFM measurements and nanomechanical measurements is in general a fantastic one but may not be appropriate to the DNTT derivatives studied in the current work because of their molecular orientation in the lattice combined with the theory's assumptions. Reasons for this are enumerated below.

The connection between electronic properties and mechanical properties in molecular semiconductors that Northrup suggested in *Appl. Phys. Lett.* 99, 062111 (2011) looks at correlations in 2D, *in the direction of charge transport and perpendicular to the faces of*

molecules. This is the direction of pi-stacking in the xy-plane in our current case. The out-of-plane z-direction sees the DNTT molecular derivatives end-on. In Northrup's acoustic deformation potential model, two quantities namely the elastic modulus E , and the charge carrier confinement length L_{eff} , are both directly proportional to the mobility μ . The transport regime considered in the deformation potential theory is the band regime which needs the charge carriers to be able to delocalize over multiple molecular length scales (in molecular crystals, it could be as large as a thousand molecules). While a deformation potential picture could make sense in the xy-transport plane, in the z-direction perpendicular to the main plane of transport, charge carrier transport is not expected to take place within a band regime. This is because the conjugated cores of the molecules along the z-direction are weakly interacting, (i) due to insulating alkyl side chains for C8-DNTT and, (ii) because of the peculiar crystal cell which hinder any overlap of the conjugated cores. In such a case, the preferential transport regime along the z-direction is likely to take place within the hopping regime. In summary, L_{eff} cannot be defined for transport along the z-direction because the transport along that direction takes place within the hopping regime. So even though the mobility and modulus may be measured along that direction using the AFM, the guideline model that relates the two quantities will unfortunately not be applicable.

The above said, the suggestion to incorporate SCLC/C-AFM measurements in tandem with nanomechanical measurements within molecular systems, appropriately oriented to expose the pi-stacking direction along the z-axis, is one we've taken seriously. Our laboratory is currently in the process of purchasing a new C-AFM module. Using this kit, we will explore nanoelectrical-nanomechanical correlations in select molecular materials in forthcoming work.

Indeed, as part of the outlook in our work, we showed that the calculated in-plane modulus from MD + DFT is proportional to the in-plane carrier mobility, suggesting consonance with the predications of Northrup's deformation potential theory. In future, one will have to identify and study molecular systems where the theory's stipulations on the fast transport direction are aligned with AFM's measurement direction, i.e., along the z-axis and normal to the thin film in the current measurement configuration.

2. I'm also wondering if there is anyway to extract two different values for the stiffness. One for the aromatic core and one for the alkyl region. I'm naively wondering if as the tip approaches the surface, one could first get the alkyl region value and then the stiff of the aromatic core as the tip approaches the surface more.

What the reviewer points out should in principle be possible, but it would require some modelling of how the ensemble of the molecules respond to the presence of the tip, and from this extract information on how compression of one part of the molecule results in compression of the second part. This is a formidable task that by itself would justify another comprehensive research paper. By just looking at the stiffness as a function of applied load using the measured data is in our view not sufficient to allow us to conclude how different parts of the molecule respond to the applied load.

In closing, the authors are very thankful for the supportive and thought-provoking comments of Reviewer #2. It has been a fantastic opportunity to be able to evolve our research article through pleasant and constructive dialogue during this review process.

Reviewer #3 (Remarks to the Author):

The current manuscript, revised by Hwang, deepens the understanding of the nanomechanical properties influenced by the hierarchical structure, through artifact-free AFM technique. The comprehensive characterization and analysis strengthen the significance and reliability of the research, with the systematic analysis contributing to a better understanding of the relationship between stiffness and charge carrier mobility. The systematic analysis contributes to the understand the relationship between the stiffness and charge carrier mobility. There are still several issues that need to be addressed.

1. The authors performed GIXRD measurements on DNTT and its derivatives. Do they have any corresponding 2D diffraction patterns or additional data? In fact, the diffraction results provide a statistical analysis of crystalline stacking in regions exposed to X-ray irradiation. While MD and DFT simulations are included for comparison, the relationship between molecular stacking orientation and mechanical properties needs clearer explanation and further clarification.

In addition to the in-plane and out-of-plane GIXRD measurements that we showed in the reply to the reviewers after the first review round and in the main manuscript, 2D diffraction patterns and grazing incidence diffraction measurements on DNTT, C8-DNTT, and diPh-DNTT thin films were published in the supporting information section of a few recent papers of ours. See A. Wittmann *et al.*, Phys. Rev. Lett. 124, 027204 (2020) where we clearly showed both in-plane and out-of-plane ordering within the crystallites of these materials, in evaporated thin films up to 50 nm, like the films studied in our work. Additional GIWAXS data were shown in the supporting information of E. Selezneva *et al.*, Adv. Mater. 33, 2008708 (2021), where we demonstrated that for films under 100 nm thickness, well ordered crystallites with the expected crystallography are guaranteed. The supporting crystallographic information files of C8-DNTT were published in our paper, G. Schweicher *et al.*, Adv. Mater. 31, 1902407 (2019) where we looked at charge transport in these evaporated polycrystalline thin films. The crystallographic structures of the chiral molecular systems, S- and R- DNTT, were solved in a recent paper, M. Volpi *et al.*, Adv. Sci. 10, 2301914 (2023). Our current work on the nanomechanical properties of these organic molecular systems builds upon the foundational knowledge we've presented in such articles that detailed their molecular orientation and crystallography within thin films.

In addition to our work, the early seminal papers of others such as T. Yamamoto *et al.*, JACS 129, 2224 (2007) and M. J. Kang *et al.*, ACS Appl. Mater. Interfaces 5, 2331 (2013)

also clearly showed both in-plane and out-of-plane ordering within evaporated thin films of DNTT, diPh-DNTT, and alkylated-DNTT in agreement with what we observe.

Below, we share additional data, namely measurements of real time X-ray reflectivity (XRR) as the DNTT-based films grow from a thickness of a few nm to around 20 nm.

Figure 1 Real-time X-ray reflectivity (XRR) scans during growth, offset for clarity. The film thickness d increases from yellow ($d = 4$ nm) to blue ($d \approx 20$ nm). Data for DNTT and two derivatives are shown in (a) – (c).

These additional data show how Bragg peaks and crystallinity develop within the polycrystalline thin films during growth by thermal evaporation. The data in Figure 1, taken together with our earlier articles documenting GIWAXS data referenced above, provide firm ground for the suggested molecular orientation, crystallographic structure, and long-range order within the grains of the polycrystalline films.

Lastly, since our AFM measurements are done on the nanoscale and with a resolution that probes intra-grain mechanical properties within the polycrystalline domains, we reckon a convincing connection between molecular stacking orientation, crystallography, and mechanical properties can indeed be drawn.

2. The carrier transport properties in this study were assessed through FET devices. Have the authors considered analyzing the electrical performance/carrier transport via conductive-AFM in conjunction with the nanomechanical measurements? This could provide additional insight into the electrical behavior at the molecular level.

On the collective suggestion made by the reviewers during the first round of review, our revised manuscript focuses on the measurement of nanomechanical properties of molecular semiconductors. Its correlation with transport properties is now only a soft touch, presented as an outlook to provide context for the importance of nanomechanics to organic electronics research. Within the outlook, theory calculations of molecular mechanics combined with carrier transport, both along the in-plane direction of fast transport, support the direct proportionality between stiffness and mobility.

This said, we have indeed considered performing conductive AFM measurements in conjunction with nanomechanical measurements, but as explained in our reply to

Reviewer #2, above, deformation potential theory suggests that one must probe the mechanics along the direction of fast carrier transport, i.e., along the pi-stacking direction. Since the DNTT molecular system stacks pillar-like along the z-direction, with its pi-stacking direction in-plane (xy-plane), doing conductive-AFM measurements on the current films along the z-direction will not probe transport in the direction of interest to the dictates of deformation potential theory's guideline.

We will take the reviewer's suggestion of undertaking C-AFM on appropriately oriented molecular semiconductors in future (i.e., where the z-direction involves fast transport and not just hopping), to probe both nanomechanics and the SCLC-mobility at every nanoscale pixel. We will identify appropriately oriented molecular systems where the pi-stacking has a significant component along the z-axis when doing tandem nanomechanical and nanoelectrical measurements. Such work will constitute a future research direction from our group, once we establish the C-AFM measurement technique in our laboratory next year.

3. In addition, the language expression and structural framework of the article are not very reasonable, which creates difficulties in reading. The author should revise the entire manuscript, optimizing aspects such as formatting and paragraph organization.

We have now carefully gone through the entire manuscript again and have carried out major revisions to the paragraphs and flow of the article. We have enhanced brevity, removed dead words, carried out formatting in areas where necessary, and reorganised paragraphs. Figure 3 has been reduced from the original 12 sub-panel figure to a 9 sub-panel figure to enhance the cohesiveness of the story overall. The supplementary information has absorbed the three subpanels that were moved out of Fig 3. The abstract and summary have also been revised for clarity. All major changes have been highlighted in the resubmitted version of the manuscript. Should the reviewer have any other specific suggestions, we'd be happy to incorporate them.

We thank the reviewer for offering us pointers for improvement through the two review rounds. It has been immensely gratifying to see the quality and the overall narrative of our work improve through their scientific dialogue and their suggestions.

Reviewer #4 (Remarks to the Author):

The authors present a joint experimental and theoretical investigation of the relationship between molecular structure, packing configuration, and stiffness of molecular crystals. Here, the molecules have a consistent pi-conjugated core structure, and differ by side chains appended symmetrically on the ends of the long axis of the core; the side chains range from hydrogen (unsubstituted), phenyl, and alkyl (with three variations, though the overall lengths are the same). Notably, the side chains do change the tilt of the molecular long axis with respect to the surface and direction of the mechanical force measurements.

The authors have done a commendable job in responding to the first round of reviewer

comments. Indeed, the major focus of the manuscript was changed. However, the authors have not sufficiently addressed what the “noteworthy results” of the work are. That careful measurements have been done, and subtleties that need to be considered when making the measurements are noted, is important work; however, this is not necessarily noteworthy. The outcomes of the physical measurements, and likewise of the DFT and MD simulations that are validated by experiment, are not surprising given the natures of the molecules and substitutions explored. Hence, the overall significance of the work is limited.

The data analyses are complete, and appear to be interpreted correctly and appropriately. The work does not appear to make spurious claims that go beyond what is measured / calculated. This is a solid paper, and the work is well done.

We are thankful to the reviewer for making time to review our work in this round and for their kind words.

Our article is an experimental demonstration of the tunability in the nanomechanical properties of molecular semiconductors based on changes in molecular structure via systematic alkyl sidechain substitution. Although such tunability may intuitively be expected, as the reviewer has rightly pointed out, the effect is subtle and extremely difficult to extract from nanoscale measurements due to various spurious force contributions operating on such small length scales. Only after identifying and removing these spurious contributions is the underlying molecular-scale tailoring of mechanics observable. The notable feat is then the first direct measurement of the impact that alkyl sidechain substitution has on the mechanical properties. We have heavily edited the abstract and summary of the manuscript to make the significance of the work more immediately apparent.

Response to Reviewers Comments

Reviewers' comments are shown in black, and ours are shown in purple.

Reviewer #2 (Remarks to the Author):

I believe that the authors have addressed the comments as much as they are able to at this time.

Reviewer #3 (Remarks to the Author):

The authors have made appropriate revisions. I recommend acceptance of the manuscript.

We thank the reviewers for engaging in a constructive dialogue through the review process and for being supportive of the final version.